# The Impact of Motor-Cognitive Dual-Task Training on Physical and Cognitive Functions in Parkinson’s Disease

**DOI:** 10.3390/brainsci13030437

**Published:** 2023-03-03

**Authors:** Yi Xiao, Tianmi Yang, Huifang Shang

**Affiliations:** Department of Neurology, Laboratory of Neurodegenerative Disorders, National Clinical Research Center for Geriatric, West China Hospital, Sichuan University, Chengdu 610017, China

**Keywords:** dual-task training, Parkinson’s disease, cognition, motor

## Abstract

Rehabilitation is a high-potential approach to improving physical and cognitive functions in Parkinson’s disease (PD). Dual-task training innovatively combines motor and cognitive rehabilitation in a comprehensive module. Patients perform motor and cognitive tasks at the same time in dual-task training. The previous studies of dual-task training in PD had high heterogeneity and achieved controversial results. In the current review, we aim to summarize the current evidence of the effect of dual-task training on motor and cognitive functions in PD patients to support the clinical practice of dual-task training. In addition, we also discuss the current opinions regarding the mechanism underlying the interaction between motor and cognitive training. In conclusion, dual-task training is suitable for PD patients with varied disease duration to improve their motor function. Dual-task training can improve motor symptoms, single-task gait speed, single-task steep length, balance, and objective experience of freezing of gait in PD. The improvement in cognitive function after dual-task training is mild.

## 1. Introduction

Parkinson’s disease (PD) is a neurodegenerative disease characterized by cardinal motor dysfunction (bradykinesia, resting tremor, rigidity, and postural instability) and non-motor symptoms such as cognitive impairments. Patients with PD have difficulty in daily activities and the difficulty increases with the disease progression. Although anti-Parkinson drugs can relieve motor symptoms, patients with PD suffer from gait and balance dysfunction in their daily lives during the disease’s course [1]. No disease-modified treatment is available. In addition to the above-mentioned motor symptoms, there are specific motor dysfunctions that add to the burden of disease on patients. Freezing of gait (FoG), defined as sudden and usually brief episodes of inability to produce effective stepping, is a commonly reported motor dysfunction in PD. FoG is related to a higher risk of falls and significantly decreased patients’ quality of life. As FoG shows a poor response to pharmacotherapy, other treatments such as rehabilitation have been proposed [2,3].

The prevalence of cognitive impairment is also high and increases with disease progression in patients with PD. At diagnosis, from 10% to 38% of de novo PD patients have mild cognitive impairment. After 5 years, around 14% to 28% of PD patients reached the point of dementia. Unfortunately, only symptomatic treatment is currently available for cognitive impairment and no treatment can modify the progression of cognitive decline. Preventing cognitive decline before the incidence of dementia has received increased attention in recent years [4].

A motor-cognitive task involves performing two independent tasks at the same time, for example, answering arithmetic questions while walking. The conduction of motor-cognitive tasks requires the involvement of both the motor and the cognitive systems. In patients with PD, the performance of the motor-cognitive task is decreased compared to age-, sex-, and education-matched healthy controls [5]. The dual-task cost is the deficit in task performance in the dual-task condition compared to that in the single-task condition. The dual-task cost is higher in patients with FoG compared to that in patients without FoG [6].

Rehabilitation is a potential way to improve physical and cognitive function in patients with PD [7]. Motor-cognitive dual-task training is a comprehensive rehabilitation method combining motor training and cognitive training [8]. The interaction between motor and cognitive tasks may have an extra benefit compared to single-task training [9]. A large number of studies researched the effect of dual-task training on improving gait, balance, motor symptoms, and cognition in patients with PD [10,11,12]. Several meta-analyses were conducted in the past few years but achieved controversial results [13,14,15]. In the current review, we summarize the current evidence of the physical and cognitive effect of dual-task training on patients with PD.

We searched the following online databases: the Cochrane Database of Systematic Reviews, MEDLINE, Scopus, EMBASE, and PROSPERO. The research was performed in September 2022. We used the following keywords and searched for them in the title and abstract: (dual task or dual-task or motor cognitive or motor-cognitive or cognitive-motor), and (Parkinson or Parkinson’s disease). We also searched through the references of related articles to capture other studies that met the eligibility criteria. The PICO strategy was used to conduct the study selection. The eligibility criteria were: (1) included subjects were PD patients; (2) the intervention was dual-task training, and the control was other training or single-task training or no intervention; (3) outcomes were cognitive or physical functions and performances. The exclusion criteria were: (1) articles not written in English; (2) the full article was not available.

## 2. Construction of Motor-Cognitive Dual-Task Training

The basic method to construct motor-cognitive dual-task training is to add a cognitive task to a motor task [8]. The cognitive task can be simple (arithmetic problems) or complex (combining verbal fluency tasks and reciting switching and working memory tasks). The level of difficulty of motor tasks also varied from gait training to a highly intensive HiBalance program [16]. The difficulty of the training can be unstable and increase with time as the participants adapt to a lower level of difficulty and reach a previously settled aim. In addition, the use of electronic devices adds numerous advantages to training. The motor task and cognitive task can be combined through video-based or virtual-reality-based exergames, adding fun and convenience to the training [17]. Portable exercise instruments, such as stationary bikes and sensors, move the scenario of rehabilitation from the hospital or rehabilitation center to home.

## 3. Effects of Dual-Task Training on Cognitive and Physical Function

### 3.1. Motor Symptom

Previous studies found that motor symptoms were improved after dual-task training [18,19,20]. Twenty patients were randomized into a 24-session single-modal training group (performing gait and cognitive training sequentially) or a dual-task training group (performing gait and cognitive training simultaneously). Both groups had a significantly decreased Movement Disorder Society-Unified Parkinson’s disease Rating Scale motor subscale score (a higher score indicating a more severe symptom) after 8 weeks of training. The dual-task training group had a larger decrease in scale score compared to the single-modal training group (15% vs. 8%). The improvement in motor symptoms could be preserved for at least 4 weeks [18]. Another study found that the shorter session of cognitive-cycling training (16 sessions) also showed benefits [20]. The Hoehn–Yahr stage of the included patients ranged from two to three, indicating that the dual-task training was suitable for mild to moderate PD [18,19,20]. Some studies assessed motor function outcomes using other scales. Previous studies found that performances of the Timed Up-and-Go Test and the 6-Minute Walk Test were improved after dual-task training [21].

### 3.2. Single Task Gait

Studies showed that some gait parameters in the single-task condition improved after dual-task training [16,22,23]. After 30 sessions of the HiBalance program, the training group had a significant improvement in speed and step length compared to the no-intervention group [16]. One study included seventeen patients and found that achieved walking distance increased after four weeks of training. The gait variability in the step length in the single-task condition decreased but did not reach statistical significance [23]. The cadence of the dual-task training group was not improved compared to that of the no-intervention group [16,24].

### 3.3. Dual-Task Gait

In daily life situations, the motor task is always accompanied by cognitive tasks, for instance, walking while reading signs. Dual-task walking can simulate the complex walking scenario in reality. Previous findings of the dual-task performance after dual-task training were controversial [16,18]. A RCT with one hundred participants found that the gait speed and step length in the dual-task condition was not improved [16]. However, another study included twenty patients and found increasing velocity and step length in dual-task conditions after training. The improvement in the step width and arm swing varied in different cognitive tasks [18]. The different cognitive tasks may contribute to inconsistent results.

### 3.4. Balance and Fall

Dual-task training can lead to an improvement in balance [10,11,16]. Large-sample-size RCTs found an improvement in balance assessed by the Mini-BESTest or Berg Balance Scale in the dual-task training group compared to the no-intervention group [10,16]. The results of previous studies regarding decreasing fall risk were controversial. An RCT with one hundred and twenty-one participants found that the 24-week frequency of falls did not decrease after dual-task training [25]. However, another RCT with a total of twenty-one patients found that 8 weeks of dual-task training decreased the 30-day frequency of the falls by 60% [19]. The dual-task training may decrease the risk of all groups for a short duration after training but the benefit is not persistent.

### 3.5. FoG

Several primary studies indicated a potential improvement in FoG after dual-task training [23,26,27]. A randomized, single-blinded, cross-over design trial included forty-six patients with PD and FoG. Patients received the cognitively challenging Agility Boot Camp Program (dual-task training) and education (control), sequentially. Compared to education, dual-task training showed a significant improvement in the subjective experience of FoG assessed by the Freezing of Gait Questionnaire. But there was no significant improvement in the objective measurement of FoG [27]. An improvement in the Freezing of Gait Questionnaire score was also found after dual-task training in an RCT that included sixty participants [21]. Another study included seventeen patients with PD and also found a decrease in scores on the Freezing of Gait Questionnaire after dual-task training, but this decrease did not reach statistical significance [23]. Killane et al. included thirteen patients with PD and FoG and found the mean number of FOG episodes per trial for the dual tasks decreased from three to one after dual-task training [26].

### 3.6. Cognitive Function

The effects of dual-task training on cognition in PD have been rarely researched [28,29]. An RCT found that the dual-task training group had an increased performance of the cognitive task in dual-task walking compared to the no-intervention group [16]. A single-blind RCT included forty patients with PD and found that the dual-task group showed a trend but no significant improvement in the executive function test (Frontal Assessment Battery and Trail Making Test) after training compared to the single-task training group (gait training) [28]. The trend of improvement in the cognitive test (Stroop) after dual-task training was also found in another study, but this improvement failed to reach statistical significance [29].

## 4. Advantages and Enhancement of Dual-Task Training

### 4.1. Dual-Task vs. Single-Task Training

Dual-task training may have some advantages over single-motor training but further studies are needed because the current evidence is not adequate. An RCT found that the dual-task training group had better gait velocity and stride length compared to the single-motor training group [28]. However, another small-sample-size RCT found that there were no differences in gait and motor symptoms between the dual-task training group and the single-task training group [18]. The improvement in balance was also comparable between the two training groups [10]. One primary study indicated that the dual-task group performed better in mediolateral balance with an eye closed but performed worse in anteroposterior balance compared to the single-motor-training group [11]. However, a previous study found that dual-task training was more effective in the improvement of motor symptoms and decreased the frequency of falls in a 30-day period compared to the separate motor and cognitive training [19].

### 4.2. Enhancement of Dual-Task Training

Several studies researched combining dual-task training and other rehabilitation to boost their effectiveness. Dual-task training plus action observation and motor imagery had a better effect on motor and cognitive function [30]. The intervention group was asked to observe a task before conducting the task while the control group only conducted the task. The intervention group had greater improvements in gait, balance, and executive function compared to the control group [30]. An RCT found that transcranial direct current stimulation enhanced the effect of dual-task training. The transcranial direct current stimulation device was placed in a small bag positioned around the participant’s hips, and was active (intervention group) or sham (control group) during the gait training. The cognitive performance during the dual-task and Timed Up-and-Go Test was improved. However, the gait and bradykinesia were not enhanced by the transcranial direct current stimulation [31]. The enhancement depends on the specific type of supplementary training. A previous study that included sixty participants showed that the multi-intervention group (combining dual-task training and aquatic therapy) did not improve motor function (FoG, balance, gait, and motor symptoms) compared to the single-intervention group (dual-task training) in PD [21].

## 5. Mechanism of Dual-Task Training

### 5.1. Motor Automaticity

One of the potential mechanisms of the physical benefit of dual-task training is the improvement in motor automaticity after training. Motor automaticity is the ability to conduct a skilled motor task without conscious attention or executive control, and plays an important role in performing dual tasks [32,33]. Functional magnetic resonance imaging has been used to assess the brain activity associated with motor automaticity in patients with PD. Participants practiced dual tasks until they could conduct the tasks automatically and then a visual letter-counting task was added to the dual tasks. Compared to healthy controls, fewer patients with PD automatically performed the combined tasks. The neuroimaging showed that only the bilateral superior parietal lobes and left insular cortex were less activated in PD when the proficiency of the task reached the automatic stage, while activities in more areas decreased in healthy controls [34]. In the automatic stage, the parietal cortex, premotor area, cerebellum, precuneus, and prefrontal cortex were more active in patients with PD compared to healthy controls [34]. The impaired motor automaticity in PD contributes to automaticity-associated motor deficits, such as akinesia, slowness of simple repetitive movements, shorter stride length, and FoG [35].

The striatum is the main pathological region in PD and is also related to the maintenance of the motor automaticity in PD [35,36]. Increased activity of the anterior putamen was found in patients with PD in the automatic stage compared to the controls [35]. Participants were asked to direct their attention back to the previously practiced automated tasks. Patients with PD had a decreased connectivity from the putamen to the motor cortex in the controlling attention stage compared to the automatic stage. However, in healthy controls, the activity and connectivity of the striatum at the controlling attention stage were comparable to those at the automatic stage [36]. Impaired sensorimotor striatum in PD results in the loss of previously learned automatic skills and the deficit in learning new skills [35].

Previous studies indicated that exercise can improve the function of the striatum. Positron emission tomography showed that expression of the striatal dopamine D2 receptor increased after high-intensity treadmill exercise in MPTP mouse models of PD compared to that in controls [37]. The improvements in striatum function were replicated in vivo in patients with PD. Four patients with PD were randomized into 24 sessions of an intensive treadmill exercise group or a control group. Increased dopamine D2 receptor binding potential and improved postural control were found in the exercise group after training, while there were no differences found in the control group [38]. Fifty-six patients with PD were randomized into an aerobic exercise group or stretching group and received six months of intervention. The performance of the oculomotor cognitive control task was improved in the exercise group compared to that of the stretching group. Functional magnetic resonance imaging found that the anterior putamen had increased functional connectivity with the sensorimotor cortex compared to the posterior putamen in the exercise group, while this phenomenon was absent in the stretching group [39]. Another study included ninety-one patients with PD and randomized them into the HiBalance group or the active control group (speech training). After 20 sessions of training, the HiBalance group had increased left putamen volumes and stronger thalamic–cerebellar connectivity in structural covariance networks, while the active control group showed no changes post-training [40].

### 5.2. Dual-Task Practice Advantage

In addition to the improvements achieved by motor training, there are additional benefits from combining motor and cognitive training, named the dual-task practice advantage. The dual-task practice advantage is described as an advantage in dual-task performance after dual-task practice compared to single-task practice (where tasks are completed sequentially) [41]. The dual-task practice advantage was found in a previous RCT of older adults. Older adults with balance problems were assigned to a single-task training group or dual-task training group (fixed-priority or variable-priority instructions). After four weeks of training, all the groups showed improvements in balance and single-task walking speed. However, only the dual-task training group had improved dual-task walking speed. In addition, the dual-task training group with variable-priority instructions was the only group that quickly showed improvement in the second week of training and maintained the training effect in the twelfth week after training [42].

The classical theoretical mechanism of the dual-task practice advantage is the allocation and scheduling hypothesis [41]. This hypothesis assumes that dual-task training improves the cognitive resource for allocation and scheduling tasks during comprehensive tasks. In a previous study, participants were arranged to conduct dual tasks, such as copying different sentences while reading stories, and then were tested for the meaning of the sentences. As they understood the meaning of sentences, the research considered that the completion of the two tasks was not automatically performed as one task but by switching between different tasks. In addition, they found that the skill to switch attention during dual tasks could be improved with practice [43]. Through dual-task training, patients can improve their ability to manage multiple tasks and the achievements can be transferred to other untrained dual tasks shared with the same neural circuit. A previous study divided dual-task performance-matched participants into hybrid training (combine single-task and dual-task training) or single-task training groups. A visual-manual task and an auditory-vocal task were trained. After training, the hybrid training group was found to have better performance in practiced dual tasks and new dual tasks compared to the single-task training group [44]. Another study explored the synergistic effect of cognitive training and motor training. A total of one hundred and thirty-six older adults were randomized into four intervention groups: cognitive training plus motor training, cognitive control plus motor training, cognitive training plus motor control, and cognitive control plus motor control. The task-set cost is the ratio between the trial mixed with two different single tasks and the trial composed of pure single tasks, and reflects the ability to memorize specific requirements of different single tasks and respond to distinct single tasks. The cognitive training was associated with a reduction in the dual-task cost. The dual-task training group predicted a decreased task-set cost, which indicated that dual-task training improved the ability to manage and switch between different tasks [45].

The neuroimaging studies found increased connectivity in older adults after dual-task training, which supported the theory. The right cerebellar vermis, left lobule V of the cerebellar anterior lobe, and precuneus were activated in the dual-task processing. The dual-task training increased the functional connectivity between these cerebellar regions and cerebellar areas related to motor and cognitive control during dual tasks [46]. However, the existence of dual-task practice advantages in patients with PD remains unclear. One study found that compared to the sequential gait and cognitive training group, the simultaneous training group had better upper extremity performance [18]. However, another study evaluated the dual-task practice advantages in PD and found that the improvements in gait parameters were comparable between simultaneous motor-cognitive training and sequential motor-cognitive training [47]. Different domains of cognitive training may contribute to the reverse results of the previous studies, as the former study trained verbal fluency, working memory, discrimination and decision making, mental tracking, and reaction time, while the latter study trained language, memory, executive function, and attention. More studies are needed in the future to establish the theory of additional improvements resulting from dual-task training in PD.

### 5.3. Guided Plasticity Facilitation Hypothesis

The guided plasticity facilitation hypothesis may contribute to the improvement in cognition by motor-cognitive dual-task training. In this hypothesis, motor training facilitates the neuroplasticity and cognitive training guides the neuroplasticity. An experiment in mice found that exercise can stimulate hippocampal neurogenesis. Mouse undergoing exercise and cognitive stimuli (environmental enrichment) recorded a greater increase in new neurons in the dentate gyrus compared to that in mice with single exercise or cognitive stimuli [48].

Brain-derived neurotrophic factor (BDNF) is a potential factor that mediates the neurogenesis induced by exercise. BDNF is known as a synapse regulator and plays an important role in the underlying mechanism of learning and memory, and hippocampal long-term potentiation (synaptic efficacy enhancement) [49]. In patients with PD, the level of serum BDNF was found to be decreased compared to that of the healthy controls, and was related to the binding ratio of the presynaptic dopamine transporter in caudate and putamen [50]. Previous animal experiments found that the level of BDNF in the circulation increases after intensive exercise [51]. A meta-analysis confirmed that the level of serum BDNF significantly increased after exercise in patients with PD [52]. In addition, the volume of the left hippocampal subfields CA1, CA4/dentate gyrus, and subiculum showed a group-dependent increase in PD [53]. When BDNF function was inhibited using a specific immune-adhesin chimera that selectively binds BDNF, the benefit of exercise on cognition vanished. The learning and recall abilities of animals that received exercise and the BDNF blocker were reduced to those of the control level [54]. However, the increase in BDNF induced by exercise is temporary, which indicates that the interaction of the motor and cognitive training is influenced by the execution arrangement and interval of the two tasks. The level of serum BDNF was elevated at day seven of the motor training program compared to baseline in patients with PD, but showed no changes compared to baseline in the following days of training and at the sixty-day follow-up after the end of the training [55].

The two tasks induce neurogenesis through distinct but complementary mechanisms as exercise stimulates the proliferation of precursor cells, whereas cognitive stimuli promote the survival of the newborn cells [56]. The synergic effect of motor-cognitive tasks may be based on neurotrophic factors. Ninety-five healthy young participants were divided into exercise, dual-task training, and control groups. Participants who had a greater cognitive improvement after training had a greater increase in BDNF and insulin-like growth factor-1. These participants who had greater increases in neurotrophic factors after exercise also showed a greater improvement after dual-task training compared to single-exercise training [57]. In patients with PD with mild cognitive impairment, the level of serum BDNF increased after the training of executive cognition compared to the control group [58]. Further research is needed to establish the role of cognitive training in dual-task training and the neural mechanism behind the synergic effect of motor-cognitive tasks.

## 6. Future Studies

There were some limitations of the previous studies. First, the selections of motor and cognitive tasks to build dual-task training were different between studies and this may influence the results [15]. In addition, the settings of the control group may be another source of heterogeneity. There were three main types of control group. The first was no control group. Researchers evaluated the effect of dual-task training by comparing the performance of patients before and after training [59]. The second was that the control group received no intervention or maintained daily activity [60,61]. The third was that the control group had a different activity, such as a single motor or cognitive task [62,63], or performed motor and cognitive tasks separately [47]. The third kind of setting was more likely to achieve a negative result compared to the others because the positive effect of exercise on physical function has been proved by previous studies [14]. However, the results of the three kinds of studies were combined in the previous meta-analysis. This partly explained the controversial results of the previous meta-analysis [13,14,15]. The settings of control groups in future studies should be chosen carefully and more attention should be paid to the separate training of motor and cognitive tasks.

There were some gaps in the specific outcomes of dual-task training. First, the effect on cognition in PD deserves more attention. The positive effects of dual-task training on cognition were found in older adults and patients with Alzheimer’s disease [64]. To date, few studies have researched the cognitive benefit of dual-task training in PD, and further studies are needed to determine whether dual-task training can kill two birds with one stone. [12,28,29] Second, because of the prosperity and diversity of rehabilitation methods in neurodegenerative disease, there is a trend to build comprehensive rehabilitation modules by combining different interventions to achieve better effectiveness [65]. Although the standardized mean difference of the different modules can be calculated and compared in a meta-analysis [14], the best way to establish evidence for a comparison between them is to use a randomized controlled trial. In addition, a training program with high-intensity and high-diversity tasks seems to result in a greater improvement in motor and cognitive function, which has guided the direction of the design of the training program [66]. However, the safety of the comprehensive module with challenging tasks should be tested because heavy training may be beyond the physical and cognitive endurance range of patients, resulting in negative results.

Individualized rehabilitation is also important, as a previous study found that the effects of dual-task training were different among patients with different characteristics. A worse cognitive status was associated with higher dual-task costs in cognition and gait in PD. The existence of other PD non-motor symptoms was related to a higher dual-task cost in cognition [67]. A large-sample-size RCT found that a lower dual-task gait velocity and a higher cognitive performance at baseline were related to a larger improvement in the dual-task velocity after training [68]. Different strategies used in dual tasks and different learning abilities may contribute to this phenomenon. Patients with PD with mild cognitive impairment used a posture-first strategy while patients with PD with normal cognition used a posture-second strategy [67]. Evidence showed that high physical activity was related to a slower apolipoprotein ε4-related cognitive decline in PD [69]. To date, the classification of PD patients in rehabilitation clinical trials has been based on motor symptoms. Future studies should consider more clinical and genetic characteristics as stratification factors and design individualized training programs for PD patients.

There are also some limitations of the current review. First, we did not pool the results of studies because the control intervention and the construction of dual-task training varied between studies. Second, we did not include studies not written in English. Third, we included RCT studies and non-RCT studies, so the quality of studies varied.

## 7. Conclusions

In the current review, we summarize the present evidence of the impact of dual-task training on motor and cognitive function in patients with PD. The summary of the current review please check Figure 1. Significant improvement has been observed in motor symptoms, single-task gait speed, single-task step length, and balance, but the effects of dual-task training on FoG, dual-task gait, and cognition are still being researched. Trends of improvement in these fields were observed in previous studies. Large-sample-size randomized control tails of dual-task training are needed to establish the effectiveness of this training. In addition, the settings for intervention and control groups should be more specific, and more attention should be paid to the additional benefit resulting from the interaction between motor and cognitive training and the mechanism of the synergic effect.

## Figures and Tables

**Figure 1 brainsci-13-00437-f001:**
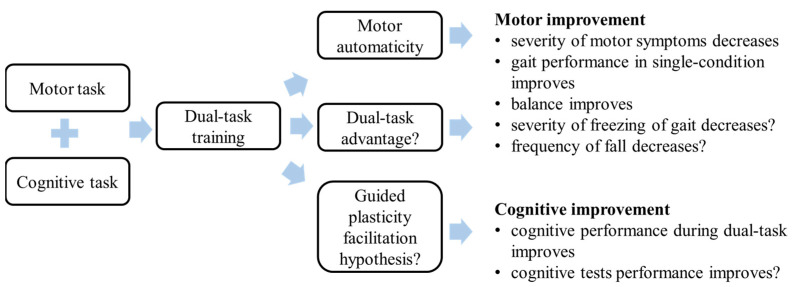
Summary of the motor and cognitive effect of the dual-task training.

## Data Availability

Not applicable.

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
