# Peer review of "The Impact of Motor-Cognitive Dual-Task Training on Physical and Cognitive Functions in Parkinson’s Disease"

_brainsci, 2023, doi:10.3390/brainsci13030437_

Round 1

Reviewer 1 Report

The current review aimed to summarize the current literature on the effect of dual-task training on motor and cognitive functions and the potential mechanism underlying the interaction between motor and cognitive training. This study is well-written and well-conducted, the authors also do a good job in the introduction and discussion reviewing the literature. There are also some minor comments that should be addressed:

The authors stated in line 16, single-ask gait speed and single-ask step length. Do the authors mean single-task!! 

In lines 16-18 the authors reported some positive effects on different motor aspects such as gait speed and balance in favor of dual-task training. They also conclude that the effect of dual-task training on other parameters and its advantage over single-task training or sequential motor and cognitive training need to be established by more studies. I think this conclusion is general and needs to be more specific. The conclusion should be clear and established in accordance with strong evidence, to prevent any misunderstanding by the readers. I suggest formalizing this conclusion again.

In addition, the aim of this study should be specified in the Abstract and authors should mention that PD patients.

The references should be cited in the text according to the journal guidelines.

I suggest the authors include a small paragraph next to the introduction to describe the search strategies, and inclusion and exclusion criteria.

Lines 58-68, this paragraph should be cited appropriately. 

Lines 173-179, and 185-186, also should be cited.

The authors stated different limitations related to eh previous studies. I suggest including some limitations related to this review. For example, this review concludes its results according to previously published studies. Pooling the results of the previous studies (meta-analysis) will provide clear evidence regarding this effect.

Author Response

To reviewer 1:

The current review aimed to summarize the current literature on the effect of dual-task training on motor and cognitive functions and the potential mechanism underlying the interaction between motor and cognitive training. This study is well-written and well-conducted, the authors also do a good job in the introduction and discussion reviewing the literature. There are also some minor comments that should be addressed:

The authors stated in line 16, single-ask gait speed and single-ask step length. Do the authors mean single-task!! 

Response:

Sorry for the mistake. It is single-task not single-ask and we corrected it in the revised manuscript.

In lines 16-18 the authors reported some positive effects on different motor aspects such as gait speed and balance in favor of dual-task training. They also conclude that the effect of dual-task training on other parameters and its advantage over single-task training or sequential motor and cognitive training need to be established by more studies. I think this conclusion is general and needs to be more specific. The conclusion should be clear and established in accordance with strong evidence, to prevent any misunderstanding by the readers. I suggest formalizing this conclusion again.

In addition, the aim of this study should be specified in the Abstract and authors should mention that PD patients.

Response:

Thanks for your advice. I formalized the conclusion and emphasized the aim and the population in the revised Abstract. The revised version is listed below.

In the current review, we aimed to summarize the current evidence of the effect of dual-task training on motor and cognitive functions in PD patients to support the clinical practice of dual-task training. In addition, we also discussed the current opinion on the mechanism underlying the interaction between motor and cognitive training. In conclusion, dual-task training is suitable for PD patients with varied disease duration to improve their motor function. Dual-task training can improve motor symptoms, single-task gait speed, single-task steep length, balance, and objective experience of freezing of gait in PD. The improvement of cognitive function after dual-task training is mild.

The references should be cited in the text according to the journal guidelines.

Response:

Thank you. We cited the references according to the journal guidelines.

I suggest the authors include a small paragraph next to the introduction to describe the search strategies, and inclusion and exclusion criteria.

Response:

Thanks for your advice. We added the search strategies. The following is the revised version.

We searched the following online database: the Cochrane Database of Systematic Reviews, MEDLINE, Scopus, EMBASE, and PROSPERO. The research was performed in September 2022. We used the following keywords and searched them in the title and abstract: (dual task or dual-task or motor cognitive or motor-cognitive or cognitive-motor), and (Parkinson or Parkinson's disease). We also searched through the reference of related articles to capture other studies that met the eligibility criteria. PICO strategy was used to conduct the study selection. The eligibility criteria: 1) included subjects were PD patients; 2) the intervention was dual-task training, and the control was other training or single-task training or no intervention; 3) outcomes were cognitive or physical functions and performances. The exclusion criteria were 1) articles not written in English; 2) the full article was not available.

Lines 58-68, this paragraph should be cited appropriately. 

Lines 173-179, and 185-186, also should be cited.

Response:

We added the references correctly in the revised version.

The authors stated different limitations related to eh previous studies. I suggest including some limitations related to this review. For example, this review concludes its results according to previously published studies. Pooling the results of the previous studies (meta-analysis) will provide clear evidence regarding this effect.

Response:

Thanks for your advice. We added the limitation. The following is the revised version.

There were also some limitations of the current review. First, we did not pool the results of studies because the control intervention and the construction of dual-task training varied between studies. Second, we did not include studies not written in English. Third, we included RCT studies and non-RCT studies so the quality of studies varied.

Reviewer 2 Report

This review tries to analyze the effects of motor-cognitive dual task training on physical and cognitive functions in Parkinson's disease (PD). This topic is important for research and clinical purposes.

Paper is well written and covers most important aspect of the topic.

I think that to clarify authors intent is useful a more detailed description of the aims of the work. Then, I think that a section of "material and methods" is needed to specificy the methodology employed in the review (i.e. time span, type of works considered,...)

Author Response

To reviewer 2:

This review tries to analyze the effects of motor-cognitive dual task training on physical and cognitive functions in Parkinson's disease (PD). This topic is important for research and clinical purposes.

Paper is well written and covers most important aspect of the topic.

I think that to clarify authors intent is useful a more detailed description of the aims of the work. Then, I think that a section of "material and methods" is needed to specificy the methodology employed in the review (i.e. time span, type of works considered,...)

Response:

Thanks for your advice. We added the search strategies. The following is the revised version.

We searched the following online database: the Cochrane Database of Systematic Reviews, MEDLINE, Scopus, EMBASE, and PROSPERO. The research was performed in September 2022. We used the following keywords and searched them in the title and abstract: (dual task or dual-task or motor cognitive or motor-cognitive or cognitive-motor), and (Parkinson or Parkinson's disease). We also searched through the reference of related articles to capture other studies that met the eligibility criteria. PICO strategy was used to conduct the study selection. The eligibility criteria: 1) included subjects were PD patients; 2) the intervention was dual-task training, and the control was other training or single-task training or no intervention; 3) outcomes were cognitive or physical functions and performances. The exclusion criteria were 1) articles not written in English; 2) the full article was not available.